# Autonomous Machine Learning Workloads on Edge Devices

## A Case Study

FirstName Surname[†]
Department Name
Institution/University Name
City State Country
email@email.com

FirstName Surname
Department Name
Institution/University Name
City State Country
email@email.com

## ABSTRACT

By 2022 more than 20% of Internet of Things (IoT) endpoint devices are expected to have autonomous Machine Learning (A-ML) capabilities [1]. The A-ML capabilities allow IoT edge devices to efficiently (by applying Machine Learning) and autonomously (without cloud compute assistance) process local information and respond accordingly in various scenarios. Moreover, employing A-ML reduces power consumption and response time related to the edge device's communication with the cloud. In this paper, we demonstrate the feasibility of integrating A-ML inference tasks on memory and performance limited (low cost) Micro Control Units (MCUs). The paper presents design tradeoffs based on quantitative performance analysis on existing low cost products.

## CCS CONCEPTS

•Computing methodologies•Hardware~Emerging technologies~Analysis and design of emerging devices and systems~Emerging tools and methodologies•Computer systems organization~Embedded and cyber-physical systems~Embedded systems~Embedded hardware•Computing methodologies~Machine learning

## KEYWORDS

Machine Learning, edge inference, CMSIS-NN, Deep Neural Networks, ARM Cortex-M, low-power computing.

## 1 Introduction

The principle barriers preventing widely distributed machine intelligence in low cost Internet of Things (IoT) edge devices are the constraints on Micro Control Unit (MCU) processing resources and power budget (i.e. Million Instructions per second (MIPS) and amount of on-chip memory).

Current research on novel Machine Learning (ML) specific HW acceleration addresses these power aware Autonomous-ML (A-ML) challenges. Novel Application Specific Integrated Circuit (ASIC) architectures include custom analog circuits [2]–[4] and computation-in-memory (CIM) architectures [5], [6] that explore the limits of the flexibility-efficiency tradeoff.

New products are also addressing these challenges in the present. Lately ARM has proposed a general-purpose micro Neural Processing Unit (microNPU) based on Helium vector-processing technology [7] and Syntiant has introduced an Audio Neural Decision Processor boasting sub-mW always-on power [8].
These A-ML proposed architectures require major silicon changes and one interesting question is can we do more with our existing low cost MCUs? In this case study we evaluate the effect of executing A-ML workloads on existing low cost, off-the-shelf MCU platforms.

For the sake of this analysis, we distinguish between two major classes of A-ML operation scenarios:

1. Time critical event processing (less than sub-second response time applications).
2. Sustained (long term) system/state classification – i.e. detect a transition (state/event) and generate a known, non-time critical event (A-ML event).

A generated A-ML event in a low-cost MCU (performance and memory constrained MCU) may trigger:
• A high-performance auxiliary MCU (class 1).
• A ML cloud server.
• A local alert or indication.

In this paper we explore design tradeoffs for class 2 applications, that is, applications that are required to generate non-time critical A-ML events. To prove feasibility of implementing such applications we benchmark runtime and power consumption of the CIFAR-10 Neural Network (NN) model, aimed at image classification. While image classification may not be required for many IoT edge devices. We will use this model as a representative upper bound on memory requirements, compute resources and classification complexity. For the sake of simplicity, we will also restrict the method for model memory footprint reduction to simple binary quantization of the original model's parameters.

Design of a specific A-ML solution requires hardware-software co-design awareness - the A-ML algorithm implementation is tailored to fit the platforms dimensions (i.e. embedded memory footprint, CPU performance and power consumption restrictions). In practice, most of the existing low-end MCU solutions are not employed today for complex A-ML applications. The main inhibiting factors are:

1. Many variants of off-the-shelf A-ML software frameworks (CMSIS-NN, uTensor, cube.ai, EdgeML+ ELL), none of which generically support small platform dimensions out of the box.
2. Lack of a ML Software Development Kit (SDK) to smoothly integrate A-ML on user platforms.
3. Lack of consistent A-ML benchmarking methodology to reflect the expected performance on specific platforms/HW architecture.

These challenges are shared by many embedded platforms (STM [9], NXP [10], etc.) and new benchmarking methodologies are being developed [11,13].

This paper is divided to the following sections:
**A-ML Benchmarking Methodology** - describes the selection of platforms and design of benchmarking methods.
**Analysis Results** – presents results and key observations.
**Discussion** – presents a summary of guidelines for current and next generation edge devices

## 2 A-ML Benchmarking Methodology

In this paper the common CMSIS-NN framework was selected as a baseline for A-ML benchmarking methodology. This framework does not require a specific Integrated Development Environment (IDE) or platform, and supports Cortex-M processors.

The Cortex-M4 and M4F based solutions were selected since vectored Multiply Accumulate Command (MAC) support is essential for high performance and power efficient implementations of A-ML inference engines (The Cortex-M0 and M3 do not support MAC).
The following low cost and performance platforms were selected: CM4F-SMALL, CM4F-LARGE, and CM4-LARGE, where F stands for floating point support and SMALL/LARGE refers to the relative SRAM size. Architectures that do not support byte level instructions or ARM instruction set were excluded from analysis since they required major adjustments to the CMSIS framework. These platforms were optimized for cost and power efficiency, not ML workload performance. As such, we were required to remove identifying information before releasing our results to the technical community. For a summary of the platform characteristics, see **Table 1.**

The pre-quantized 8 bit fixed-point CIFAR-10's model was generated based on the CMSIS_5-5.6.0 framework (The model code was generated with minor adjustments to the framework in order to support missing macros and compiler intrinsics). More advanced techniques such as model pruning and ternaryvalue representation were not employed.

All benchmarks were developed with GNU v9.2.1 (Linaro) compiler; All models reuse the same CMSIS-NN static library optimized for Cortex M4 processors and optimized for area (-Os).

The CIFAR-10 inference model[12] is considered a high-end benchmark for MCUs since it receives as input three channels (R,G,B) of a 32-by-32 image, and the inference task consists of correctly classifying the image as a member of one out of ten categories (bird, cat, dog, airplane, etc.). The amount of memory required to hold the model parameters and the computational complexity of each of the model's layers is described in **Table 2**.

The number of cycles it takes to execute each layer and the entire inference model were measured eight times via clock cycle event counter breakpoints. Computation was completely deterministic and showed no computational instability or cold start effects. Clock division factors and correct clock frequency were verified by examining the CSTL1 register and Runtime Object Viewer (ROV).

Average power was measured eight times on platforms that support built-in power measurement (CM4F-SMALL, CM4F-LARGE). The CM4F-SMALL model parameters could not fit into SRAM and were placed on FLASH. Other platforms performance was measured both on FLASH and SRAM. The absolute runtime was calculated by multiplying the cycles by the MCU clock period. Other standard figures of merit include: inferences per second (the inverse of the model's runtime) and GOPS/W (Giga operations per second per Watt).

Experimental Setup

| Normalized System Parameters | Edge Device Type | | |
|---|---|---|---|
| | CM4F-SMALL | CM4F-LARGE | CM4-LARGE |
| Processor | Cortex-M4F | Cortex-M4F | Cortex-M4 |
| Frequency | 1 | 1.0 × | 1.6 × |
| SRAM | 1 | 3.2 × | 3.2 × |
| Flash | 1 | 5.8 × | 2.9 × |
| Cache | 1 | - | - |
| Unit Price | 1 | 2.2 × | 1.4 × |

**Table 1: Comparison of the system parameters of the edge devices selected for benchmarking compared to the CM4F-SMALL baseline. Cortex-M4F CPUs contain a Floating Point Unit (FPU) and a Memory Protection Unit (MPU). No floating point operations are used in the benchmark.**

CIFAR-10 Model Parameters

| Layer | Memory[B] | MACs |
|-------|-----------|------|
| conv1 | 2432 | 2.46E+6 |
| relu1 | 0 | 1.64E+4 |
| pool1 | 0 | 3.69E+4 |
| conv2 | 12816 | 3.28E+6 |
| relu2 | 0 | 2.05E+3 |
| pool2 | 0 | 4.61E+3 |
| conv3 | 12832 | 8.19E+5 |
| relu3 | 0 | 1.02E+3 |
| pool3 | 0 | 1.02E+3 |
| ip1 | 5130 | 5.12E+3 |
| softmax | 0 | 1.50E+1 |

**Table 2: The required memory in bytes (parameter weights & bias) and theoretical number of multiply – accumulate (MACs), operations. Each pair of non-MAC operations is counted as a single MAC operation**

## 3 Analysis Results

### 3.1 Cycle-by-Cycle Analysis

A Cycle-by-Cycle methodology was utilized to analyze the number of cycles it takes for a single inference to complete. The analysis outcomes (demonstrated in Fig. 1) show that the memory space limited CM4F-SMALL (FLASH based) edge device would have superior performance compared to SRAM based edge devices, and by proxy - superior power consumption (this contradicts the SRAM based fast memory access superiority assumption). The CM4F-SMALL performance boost is related to a tightly connected 8-KB 4-way random replacement cache that minimizes the A-ML active power consumption and reduces the number of wait states. In the CM4F-LARGE and CM4-LARGE edge devices, though SRAM access is faster than FLASH, the SRAM configuration is such that both instruction fetch and read access of model parameters compete for access to the SRAM.

### 3.2 Energy Efficiency Analysis

The energy efficiency power measurement analysis demonstrated that although the CM4F-SMALL is faster than the CM4F-LARGE, it also consumes more power for this application (24.482mW compared to 18.182mW). The relation between the performance (measured in inferences per second) and energy efficiency measured by GOPS/W (Giga-operations per second, per Watt) is presented in Fig. 2. The analysis shows that when both systems execute from flash they have similar energy

efficiency, i.e. battery life on both systems is more or less the same.

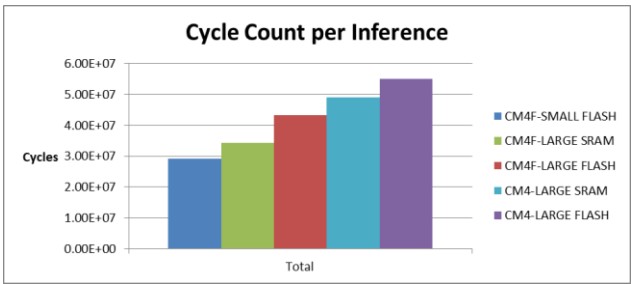

**Figure 1: Inference time measured in cycles for each test configuration. The location of model parameters is explicitly stated as SRAM or FLASH accordingly.**

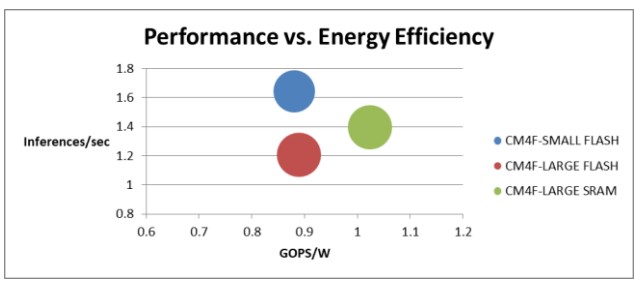

**Figure 2: Inference time measured in cycles for each test configuration. The location of model parameters is explicitly stated as SRAM or FLASH accordingly.**

The A-ML benchmarking methodology proposed in this paper also illustrates the CM4F-SMALL user experience and performance superiority. The standard performance benchmarks do not capture this distinction, as the Coremark an Dhrystone scores for both edge devices are comparable (performance presented in the datasheets are: 1.25 and 1.22 DMIPS/Mhz, and 3.083 and 3.41 Coremark/Mhz respectively).

### 3.3 Additional Performance Acceleration Provided by Local SRAM

The paper analysis also compares the relative performance speed-up from storing model parameters in SRAM compared to FLASH in the CM4F-LARGE and CM4-LARGE edge devices (see Fig. 3); the CM4F-LARGE exhibits improved performance on all layer operations, however the CM4-LARGE does not.

A possible explanation for this behavior is due to additional wait states inserted by SRAM memory access arbitration between instruction and data access.

### 3.4 Inference Response Time Analysis

**Figure. 4** demonstrates the inference response time analysis results. The analysis shows that if the most important criterion is maximum absolute performance, the CM4F-SMALL, which has model parameters stored on FLASH, exhibits performance comparable to that of the CM4-LARGE with model parameters stored on SRAM.

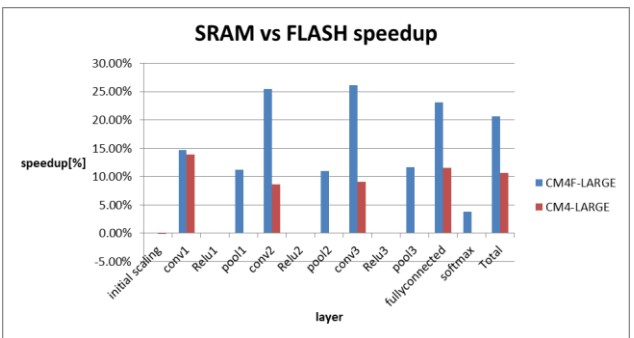

**Figure 3: Relative speedup for each device, between configuration with model storage on SRAM and that of model storage on FLASH.**

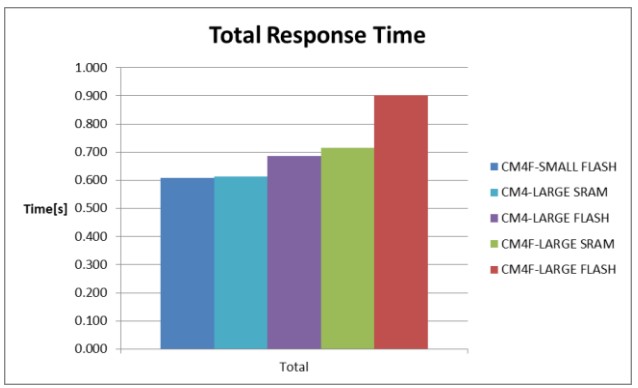

**Figure 4: Total response time measured in seconds from the moment a new image is available to the time it has been successfully classified. The location of model parameters is explicitly stated as SRAM or FLASH accordingly.**

### 4 Discussion

The analysis in this paper demonstrated that very large A-ML complex models (i.e. CIFAR-10) can be implemented on low cost off the shelf MCUs within reasonable performance for class 2 applications. This class of models can easily be added to embedded platform offerings and add additional opportunities for Machine Learning based use cases.

### 3.5 Power Consumption Profile

**Figure. 5** presents a representative power consumption profile over time for a convolutional neural network. The profile clearly has two phases during operation. This means that accurate power measurements require multiple samples and that the sampling interval should not fall solely on the minimum or maximum value

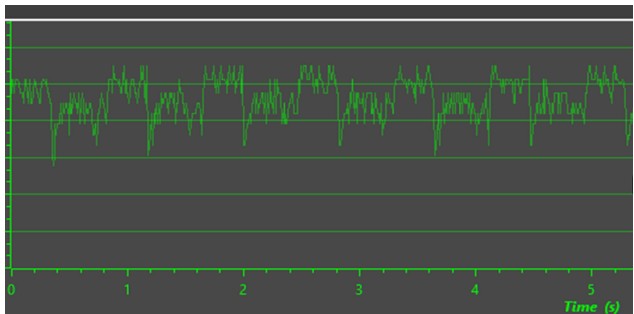

**Figure 5: Power consumption vs. time on multiple inference iterations**

A smart selection of cache based products can also boost performance (both power efficiency and response time) considerably. One step further in performance improvement can be achieved by FLASH-RAM partitioning optimization and increasing CPU clock frequency.

The performance impact of the edge device selection and memory partitioning strategies cannot be quantified by traditional benchmarking (i.e. Coremark, Dhrystone) therefore, new A-ML specific benchmarks, as demonstrated in this paper need to be applied.

An interesting area of investigation outside the scope of this paper is an end-to-end analysis of power consumption when taking into account communication patterns. For example, an anomaly detection application may be required to communicate a state change only when a low probability anomaly occurs. In this scenario power consumption will be much lower than that of a cloud based compute system which is required to continuously transfer large amounts of data.

Additional performance improvements and wider use of A-ML solutions can be achieved by:

1. Tuning next generation architecture to support A-ML needs.
2. Providing SDK, maintenance and debugging features.
3. Developing an integrated A-ML benchmarking environment.

## ACKNOWLEDGMENTS

I would like to thank my manager and department manager for encouraging me to be curious and explore beyond the horizons of my job function responsibilities.

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
