# OpenReview forum: "Autonomous Machine Learning Workloads on Edge Devices: A Case Study"
_tinyml.org/tinyML/2021/Research_Symposium — Reject_

### Official Review · AnonReviewer2 · 2021-01-07

**Overall Merit Score:** 2

**Brief Summary:**

The paper presents a performance and power benchmark for inference of a neural network designed for the CFAR-10 data set.
It compares three deeply embedded hardware platforms with Cortex-M4 processors but different memory architectures and sizes.
In particular the impact of parameter storage in SRAM or Flash and the impact of a data cache is discussed.
CMSIS-NN is used as a mapping framework. A simple binary quantization is used for model footprint reduction.

**Detailed Comments:**

The paper presents interesting measurements about neural network inference on state of the art MCUs.
It is suited to point out potential relevance of data caches for some applications.
It is not suited to derive conclusions for future MCU architectures.

**Paper Strengths:**

* The paper elaborates on the practically relevant question on what performance levels of neural network inference can be achieved in deeply embedded platforms (MCUs).
* It provides a layer-by-layer comparision of execution speed with parameters in SRAM vs. Flash.


**Paper Weaknesses:**

* State more clearly how the binarization of the model is done and how the binarized model is executed in conjunction with CMSIS-NN
* Comment on the accuracy of the binarized network. The value of the benchmark depends on the usefulness of the example.
* The model parameter table (Table 2) differes significantly from the model described in [12]. Please explain the deviations.
* A significant amount of SRAM footprint and bandwidth is required for reads and writes of activations. Please discuss how this plays into your benchmark results (e.g. caching of activations).
* Discuss how you trade cache vs. SRAM size and how different cache implementations impact performance.


**Poster (If Paper Is Rejected):**

1: Yes, ok for poster sesion to nurture work

**Reviewer Confidence:**

5: The reviewer is absolutely certain that the evaluation is correct and very familiar with the relevant literature

---

### Official Review · AnonReviewer1 · 2021-01-26

**Overall Merit Score:** 2

**Brief Summary:**

Evaluate CIFAR-10 on IoT MCU with ML accelerator, especially for some event detection(e.g. immediate state change.), not for time series data analysis

**Detailed Comments:**

The approach sounds quite interesting and promising but I want to see more progressed report of this. I don't say that the current one is not unmature but I really want to see more progressed one to be accepted.

**Paper Strengths:**

- discussed A-ML Benchmarking Methodology
- power measured via hw counter
- comparison between SRAM and Flash

**Paper Weaknesses:**

- simple binary quantization of the original model’s parameters.
- depends highly on CMSIS-NN
- power consumption data is weak, as mentioned.

**Poster (If Paper Is Rejected):**

1: No, paper is below bar for poster as well

**Reviewer Confidence:**

4: The reviewer is confident but not absolutely certain that the evaluation is correct

---

### Official Review · AnonReviewer3 · 2021-01-30

**Overall Merit Score:** 1

**Brief Summary:**

In the paper, the authors explore the design trade-offs for applications that are required to generate non-time critical A-ML events. An NN model trained on CIFAR-10 dataset is selected and three different low cost platforms, including CM4F-Small, CM4F-Large, and CM4-Large, are benchmarked and compared on both latency, energy efficiency, etc.

**Detailed Comments:**

In this paper, the authors use an NN model trained on Cifar-10 dataset to compare different hardware platforms. The paper presents some interesting experiments and observations, e.g., the performance acceleration provided by local SRAM. However, the paper suffers from several important drawbacks, including limited benchmark efforts, lack of in-depth analysis, etc. More networks and more analysis are needed to provide comprehensive analysis and give people the necessary insight. Meanwhile, the paper also suffers from bad readability. The paper is not well organized and there are many typos in the paper. The manuscript definitely needs to be improved.

**Paper Strengths:**

The paper has the following strengths:
- The comparison on the performance acceleration provided by local SRAM is interesting.
- It is interesting to find for CM4-Large, using SRAM does not improves the performance too much.


**Paper Weaknesses:**

The paper has the following major weaknesses:
•	Only a single model is used for benchmarking. More models with diverse architectures and layer characteristics are needed for more detailed comparison of different hardware platforms.
•	Lack of in-depth analysis: while the paper presented some benchmarking results on latency or energy efficiency, more in-depth analysis is needed. For example, the paper shows the impact of local SRAM on latency is small for CM4-Large. It is more interesting to provide analysis on why this is the case, e.g., whether the problem is bounded by compute or memory access and how large the arbitration between instruction and data access is in practical.
•	The paper writing could be improved. For example, what are the class 2 applications? What are the use-cases for these applications in practice? Why a simple classification network trained on CIFAR-10 is a representative workload? Also, Section 4 should be placed after section 3.5.


**Poster (If Paper Is Rejected):**

1: No, paper is below bar for poster as well

**Reviewer Confidence:**

5: The reviewer is absolutely certain that the evaluation is correct and very familiar with the relevant literature

---

### Decision · Program_Chairs · 2021-02-05

**Decision:**

Reject

**Comment:**

Thank you for your submission.

Following careful consideration by our reviewers, we regret to inform you that we are unable to accept your submission.

Please refer to the reviewer comments for your reference. We hope you find this information helpful for submission to another venue, and we hope to see more of your work in the future.